# Pyrogenic carbon contribution to tropical savanna soil carbon storage

Yong Zhou [1,2] ✉, A. Tyler Karp [3], Abigail Schmidt[1] & Corli Coetsee [4,5]

Savannas are fire-prone ecosystems that contribute substantially to global fire emissions, but these emissions may be partly offset by deposition of fire-derived, persistent pyrogenic carbon (PyC) in soils. Although estimates of PyC contributions to soil organic carbon (SOC) storage in savanna exist, factors driving its accumulation remain unclear due to limited measurements with consistent methods. To address this, we sampled 253 sites across tropical savannas in Kruger National Park, South Africa, spanning broad gradients in fire regimes, grass biomass, rainfall, and soil texture. Here we show, PyC measured with $H_2O_2/HNO_3$ digestion contributed, on average, 14.08% (se = 0.36%, $n = 253$) of SOC in surface soils, with values up to 40%. While fire frequency and grass biomass influenced soil PyC stocks, savannas with higher clay content and lower rainfall – conditions favoring PyC preservation – tended to accumulate more. These results demonstrate PyC's significant contribution to SOC storage and highlight environmental factors driving its accumulation in tropical savannas, providing an empirical basis for understanding fire's role in the savanna carbon cycle.

Fire plays a crucial role in shaping ecosystem processes[1], but also releases carbon from the biosphere to the atmosphere through the combustion of plant biomass, surface litter, and even soil organic matter[2,3]. However, fires may also serve as a carbon sink[4–6], particularly in fire-prone savannas where they are fueled by grasses that typically recover during the next growing season. Meanwhile, the deposition of pyrogenic carbon (PyC) from the incomplete combustion of grass biomass during fires has the potential to significantly enhance soil organic matter stability[7–9], as PyC is far more resistant to biological decomposition than other forms of soil organic carbon (SOC) and can persist in soils for decades to thousands of years[10]. Indeed, a recent land-surface modeling study suggested that PyC accumulation in soils could surpass fire-related carbon losses in terms of long-term carbon sequestration in tropical grasslands and savannas[5], which account for nearly 70% of the global burned area[11,12] and contribute over 80% of global fire-produced PyC (ranging from 40 to 340 million tons per year)[4,5,13].

However, not all fire-produced PyC in tropical savannas (and other ecosystems) is integrated into the soil profile where it contributes to soil organic matter formation[8,14], as PyC is highly mobile and can be physically transport by wind or runoff to various reservoirs, such as river and ocean sediments[10,15–17]. Therefore, quantifying PyC contributions to SOC is essential for understanding the legacy carbon sink created by fires in terrestrial ecosystems[5,7]; however, existing measurements are primarily from the Northern Hemisphere and in forest ecosystems[18–23] and, importantly, cannot be compared due to inconsistent methods for determining soil PyC content across studies[7,24–26]. As a result, we lack quantitative estimates of the overall contribution of PyC to SOC and a comprehensive understanding of the factors driving PyC accumulation in soils across tropical savannas.

The amount of PyC accumulated in soils is influenced by factors affecting PyC production, such as fire regimes and fuels[18,27,28], as well as factors that impact PyC preservation, including soil texture and rainfall[10,29]. Despite the substantial contribution of tropical savannas to

[1]Department of Wildland Resources and Ecology Center, Utah State University, Logan, Utah, USA. [2]Department of Ecology, Evolution, and Marine Biology, University of California, Santa Barbara, Santa Barbara, California, USA. [3]Department of the Geophysical Science, University of Chicago, Chicago, Illinois, USA. [4]School of Natural Resource Management, Nelson Mandela University, George, South Africa. [5]Scientific Services, Kruger National Park, Private Bag X402, Skukuza, South Africa. ✉e-mail: yongzhou@ucsb.edu

global PyC production, the burning of grass biomass in these ecosystems generally produces finer PyC particles[30]. These finer pyrogenic particles are more susceptible to transportation by wind or runoff[31], potentially leading to lower PyC retention at the burning site and, consequently, in the soil. This contrasts with well-studied forest ecosystems, where the combustion of woody biomass generates coarser pyrogenic particles that are more likely to remain at the burning site, with production factors, such as fire severity, substantially influencing PyC accumulation in soils[18,27]. Therefore, in savanna ecosystems, factors influencing on-site PyC preservation, such as soil texture and rainfall[10,29,32], may outweigh those affecting PyC production in determining overall PyC accumulation in soils. Disentangling the deterministic and interactive effects of these factors on soil PyC accumulation requires spatially explicit sampling across multiple sites on a large scale, spanning broad environmental gradients.

Here, we first compiled a global dataset of existing soil PyC measurements in natural ecosystems (Fig. 1) to highlight: (1) that while substantial data exist at the global scale, soil PyC measurements are scarce in tropical savannas, where frequent fires are expected to account for a significant portion of global PyC production; and (2) that inconsistent methodologies for determining PyC concentration in soils present challenges for cross-comparisons and identifying key factors driving soil PyC accumulation in tropical savannas. We then fill these gaps by measuring soil PyC across tropical savannas that span broad gradients in grass biomass, fire frequency, rainfall, and soil texture in Kruger National Park (hereafter Kruger), South Africa (Fig. 2). Soil samples were analyzed for PyC using a peroxide-weak nitric acid method[26]. The well-documented long-term vegetation, fire, and environmental datasets from Kruger, combined with the consistent methodology for soil PyC determination, enable us to (1) assess the contribution of PyC to SOC storage, and (2) identify the key drivers influencing soil PyC accumulation in tropical savannas. Finally, we discuss the implications of our results for savanna management aimed at enhancing PyC contributions to the long-term carbon sequestration potential of tropical savannas.

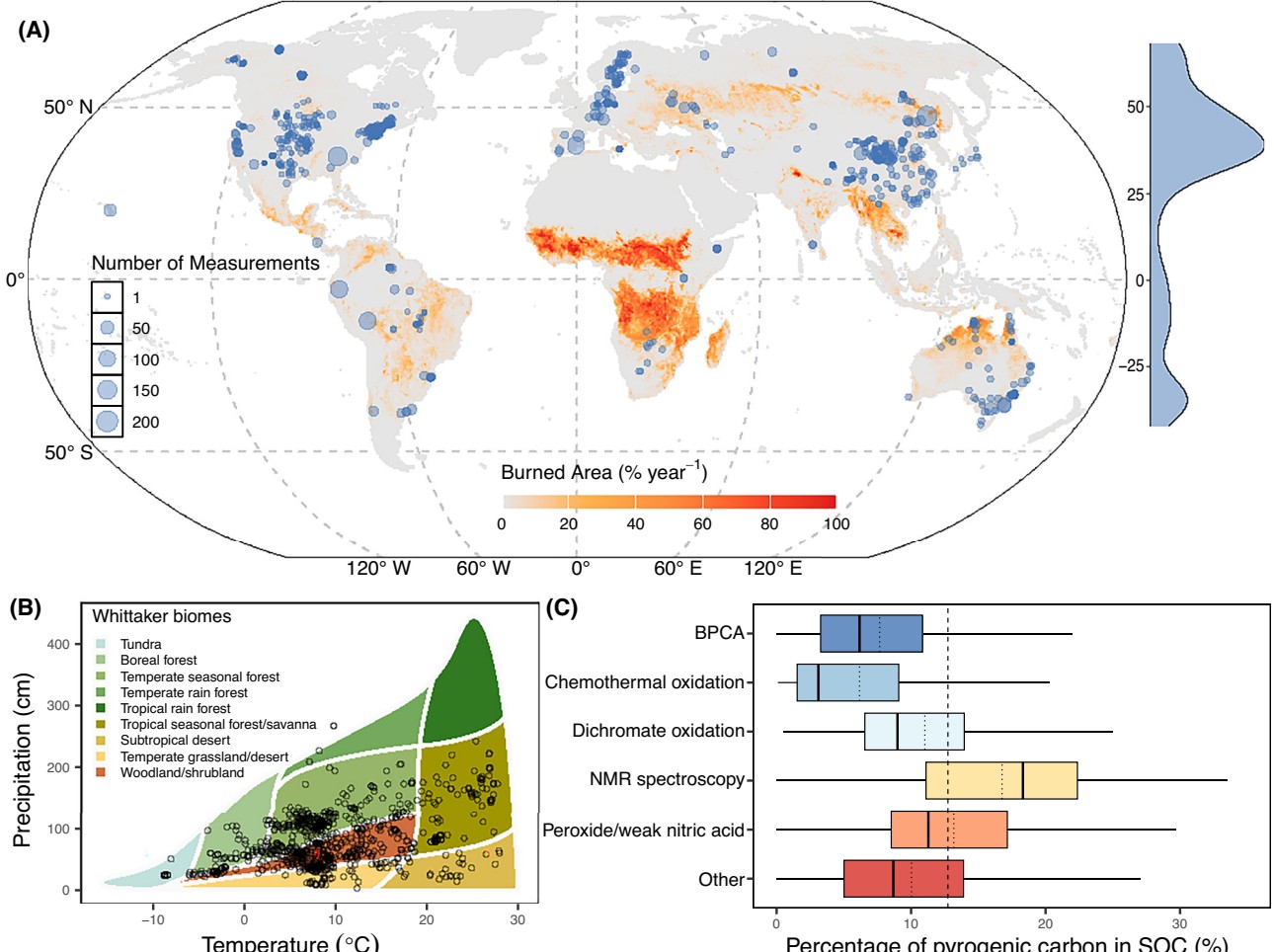

**Fig. 1 | Global distribution of soil pyrogenic carbon (PyC) measurements. A** The spatial distribution of soil PyC measurements overlaid on the global mean annual burned area. Soil PyC data were extracted from the literature (*see* Methods). The global mean annual burned area is represented as a percentage of the burnable land area per year (% year⁻¹) in each 0.25° × 0.25° grid cell from 2001 to 2020. Data were from the Global Fire Emission Database (GFED5)[12]. The base map is modified from Natural Earth. **B** The biome distribution of study locations based on mean annual temperature (°C) and mean annual rainfall (cm), following the Whittaker biome classification. **C** Boxplot of percentage of PyC in soil organic carbon (SOC) (%) according to different methodological approaches. The box represents the interquartile range from the 25th to the 75th percentile of the data, with the line inside the box indicating the median value. Whiskers extend to the highest and lowest values within 1.5 times the interquartile range. The dotted line within the box indicates the mean value, and the dashed line indicates the mean value (mean = 11.44, $n = 3456$) of the whole dataset. The methodological approaches used for PyC determination include benzenepolycarboxylic acid (BPCA) ($n = 815$), chemothermal oxidation ($n = 637$), dichromate oxidation ($n = 593$), nuclear magnetic resonance (NMR) spectroscopy ($n = 585$), peroxide/weak nitric acid ($n = 414$), and other ($n = 502$) (*see* "**Methods**"). Source data are provided as a Source Data file.

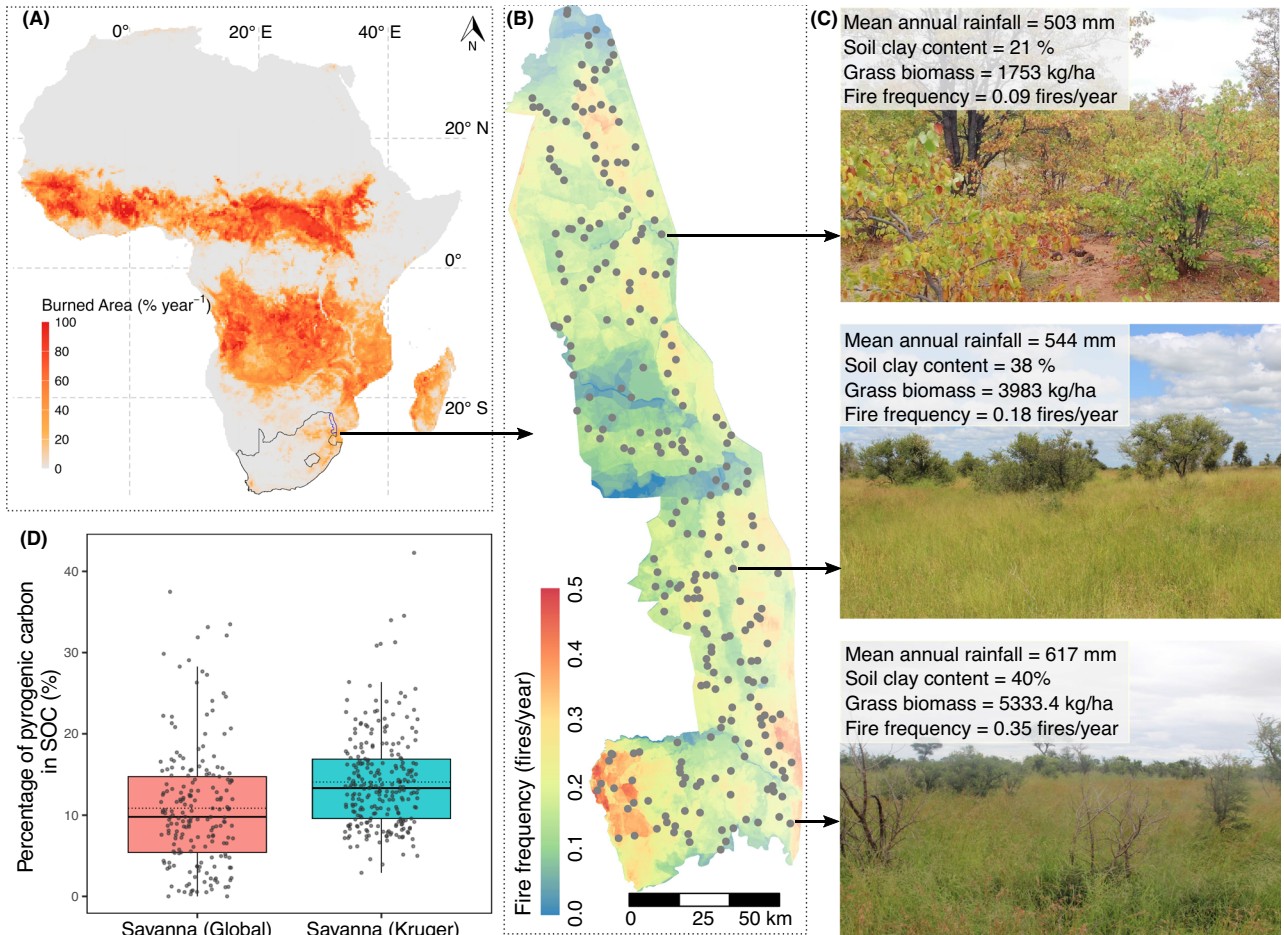

**Fig. 2 | Study sites and the contribution of pyrogenic carbon (PyC) to soil organic carbon across tropical savannas in Kruger National Park, South Africa.** **A** Geographic location of Kruger National Park within South Africa, on the African continent. The mean annual burned area is represented as a percentage of the burnable land area per year (% year⁻¹) in each 0.25° × 0.25° grid cell from 2001 to 2020. Data are from the Global Fire Emission Database (GFED5)[12]. The base map is modified from Natural Earth. **B** Spatial distribution of 253 Veld Condition Assessment (VCA) sites overlaid on a map showing fire frequency (fires/year) across tropical savannas in Kruger National Park. Fire frequency was calculated based on fire records from 1941 to 2023. The Kruger National Park shapefile was provided by South African National Parks Scientific Services. **C** Images illustrating three VCA sites, including their mean annual rainfall, soil clay content, grass biomass, and fire frequency. **D** Boxplot comparing the percentage of PyC in soil organic carbon (SOC) (%) across global savannas synthesized from the literature ($n = 197$) and savannas within Kruger National Park, South Africa ($n = 253$). The box represents the interquartile range from the 25th to the 75th percentile of the data, with the line inside the box indicating the median value. Whiskers extend to the highest and lowest values within 1.5 times the interquartile range. Individual data points are shown as jittered dots. Note that PyC measurements for global savannas were obtained from varying soil depths and determined using different methodological approaches. Source data are provided as a Source Data file.

## Results and discussion

### The global contribution of pyrogenic carbon to soil organic carbon

Through the literature synthesis, we collected 3456 PyC measurements from varying soil depths at 1139 unique locations across global natural ecosystems (Fig. 1A), covering a wide mean annual temperature (−8 to 28 °C) and precipitation (33 to 2676 mm) (Fig. 1B). However, neither climate variable significantly influenced the percentage of PyC in SOC (Supplementary Fig. 1). Soil PyC concentrations were determined using various methods that had significantly different distributions, with nuclear magnetic resonance spectroscopy yielding the highest average percentage of PyC in SOC, chemothermal oxidation yielding the lowest, and other methods falling in between (Fig. 1C). The mean value of the peroxide/weak nitric acid method used in our study was close to the grand mean across all methods (Fig. 1C). Most of these measurements were from forest ecosystems ($n = 2255$, 65% of total PyC measurements) (Fig. 1B, Supplementary Fig. 2), particularly in the northern hemisphere (Fig. 1A). Across different ecosystems with > 100 PyC measurements, the mean percentage of PyC in SOC followed the order:

desert > grassland > forest > savanna > shrub/woodland (Supplementary Fig. 2). However, these patterns varied depending on the method used to measure PyC (Supplementary Fig. 2), further highlighting the challenge of comparing results across studies due to inconsistent methodologies for PyC determination. Additionally, subsurface soils (>30 cm) had a slightly higher percentage of PyC in SOC compared to surface soils (0–30 cm) (Supplementary Fig. 3). Overall, averaged across different analytic methods, ecosystems, soil depths, PyC represents 11.44% (se = 0.17, $n = 3456$) of SOC at the global scale, which is close to an earlier synthesis[7] that reported a value of 13.7%. We note that, unlike our synthesis, this earlier work included agricultural sites, which tend to have a higher percentage of PyC in SOC compared to natural ecosystems[7].

Our synthesis confirmed that PyC measurements from tropical savannas are underrepresented in global datasets ($n = 197$; 6% of total PyC measurements, despite savannas covering ~20% of Earth's land surface) (Supplementary Fig. 2). Measurements from the African continent, which contribute significantly to the global burned area[11,12] (Fig. 1A) and are expected to produce most of the global PyC[4,5,13], were

particularly sparce ($n = 16$ in Botswana[32] and $n = 2$ in Zimbabwe[33]). Our study provides extensively sampling and measurement of soil PyC concentration across tropical savannas at a regional scale (Fig. 2B). Overall, we found that PyC contributed, on average, 14.08 % (se = 0.36 %, $n = 253$) of SOC within the surface soils (0–5 cm) (Fig. 2D). This value is slightly higher than the global savanna average reported in the literature (mean ± se = 10.85 ± 0.53%, $n = 197$) (Fig. 2D). However, we note that comparing our regional average with the global savanna average involves additional uncertainties arising from variations in soil depths and analytical methods.

## Key drivers of soil pyrogenic carbon accumulation in tropical savannas

We applied a random forest model to analyze a broad gradient of factors influencing PyC production and preservation captured in our dataset (Fig. 2C, Supplementary Fig. 4) to examine key deterministic and interactive factors driving soil PyC accumulation. Overall, soil PyC stock across tropical savannas in Kruger ranged from 0.19 to 3.45 Mg C/ha, with an average of 1.23 Mg C/ha (se = 0.05 Mg C/ha, $n = 253$). The cross-validated performance metrics from the random forest model for predicting soil PyC stock showed an $R^2$ of 0.67 and an $RMSE$ of 0.44 Mg C/ha. The spatial distribution of model residuals revealed no obvious clustering (Supplementary Fig. 5), and Moran's I of the residuals was close to zero (Moran I statistic = −0.004, $p < 0.0001$), which together indicate no evidence of spatial bias in the model predictions. For comparison, we also applied a generalized linear model (GLM) to our dataset (Supplementary Table 1). While its overall predictive performance was slightly lower ($R^2 = 0.6$) than that of the random forest, the best-fit GLM identified a similar set of key predictors influencing soil PyC stock, supporting the robustness of the relationships observed in our dataset.

The most influential factors in soil PyC accumulation across tropical savannas in Kruger were related to PyC preservation, including soil clay content and rainfall (Figs. 3, 4). Soil PyC stock increased with clay content, particularly between 20% and 40%, where PyC increased by ~ 0.03 Mg C/ha for each 1% increase in clay content (Fig. 3B). Soil PyC stock decreased with increasing mean annual rainfall, particularly from 420 to 550 mm rainfall, where PyC decreased by ~ 0.05 Mg C/ha for each 10 mm increase in rainfall (Fig. 3C). Together, savannas with

higher soil clay content and lower rainfall-conditions that favor PyC preservation-tend to accumulate more PyC in the soil (Fig. 4A). Landscape elevation also influenced soil PyC accumulation, with savannas at lower elevations accumulating more PyC compared to those at higher elevations (Fig. 3G). Landscape slope had minimal influence on soil PyC accumulation (Fig. 3I), likely due to the relatively flat terrain of savanna sites sampled across Kruger, where slopes ranged from 0.23° to 8.97°, with an average of 2.15° ($n = 253$).

To a lesser extent, soil PyC accumulation across tropical savannas in Kruger was also influenced by factors associated with PyC production, such as grass biomass and fire frequency (Figs. 3, 4). Soil PyC stock increased by approximately 0.04 Mg C/ha for every 1000 kg/ha increase in grass biomass (Fig. 3D). Soil PyC stock decreased with increasing woody cover, but only in open savannas with less than 20% woody cover (Fig. 3E). Tree-grass competition for resources such as light, water, and nutrients reduces grass biomass and fuel loads, which in turn reduces fire frequency[34] (Supplementary Fig. 6). This mechanism is consistent with our observations that soils accumulated more PyC in more open savannas with greater grass biomass and more frequent burns (Fig. 4F, Supplementary Fig. 6). Soil PyC stocks increased slightly with greater fire frequency across this tropical savanna (Fig. 3F).

Our results suggest that factors influencing PyC preservation outweigh those influencing PyC production in determining soil PyC accumulation across savannas in Kruger (Fig. 4). For example, grass biomass and fire frequency were less influential when considered together with soil clay content and mean annual rainfall (Fig. 4B–D). Savannas with soil clay content greater than 30% generally accumulated more PyC than those with less than 30% clay content, regardless of grass biomass and fire frequency (Figs. 4B, D). Similarly, savannas with a mean annual rainfall of less than 500 mm accumulated more PyC compared to those receiving more than 500 mm, also independent of grass biomass and fire frequency (Figs. 4C, E). While these results align with some studies showing that higher soil clay content enhances physical protection[35,36] and lower rainfall reduces soil erosion[17,19,37]−both promoting greater PyC accumulation−other studies also emphasize fire severity as a key factor influencing PyC accumulation in forest ecosystems[18,19]. Severe fires can produce greater amounts of PyC from standing trees and coarse woody debris, which is

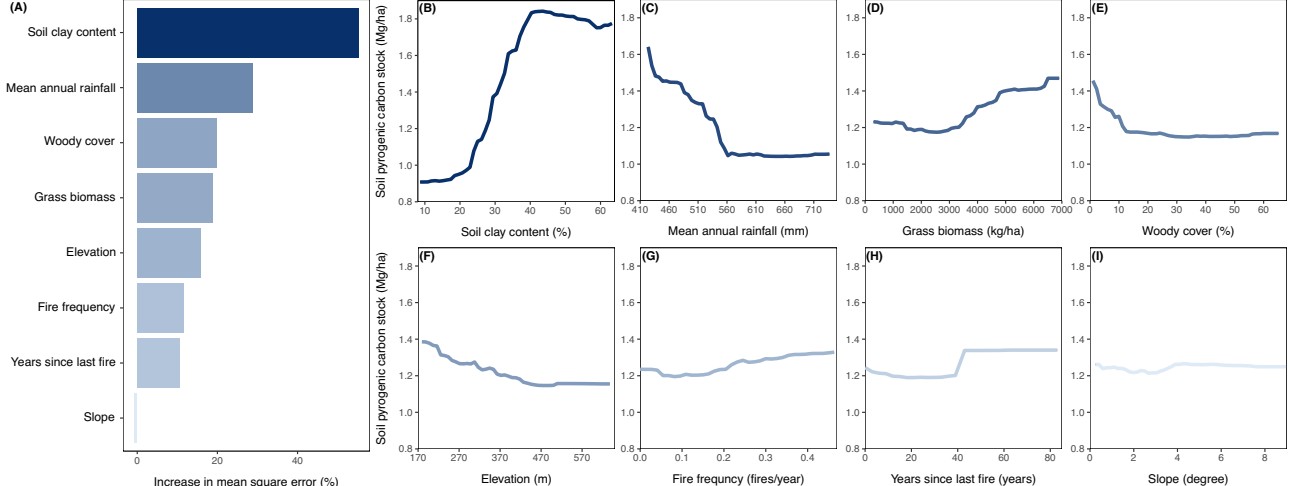

**Fig. 3 | Contribution and influence of explanatory variables on determining soil pyrogenic carbon (PyC) stock accumulation across tropical savannas in Kruger National Park, South Africa. A** The variables are ranked based on their relative importance, where an increase in mean square error reflects the change in the random forest model when each variable is omitted. Partial dependence plots

(PDPs) show the predicted soil PyC stock (Mg C/ha) as a function of each predictor variable, including soil clay and silt content (%) (**B**), mean annual rainfall (mm) (**C**), woody cover (%) (**D**), grass biomass (kg/ha) (**E**), elevation (m) (**F**), fire frequency (fires/year) (**G**), years since the last fire (years) (**H**), and slope (degrees) (**I**). Source data are provided as a Source Data file.

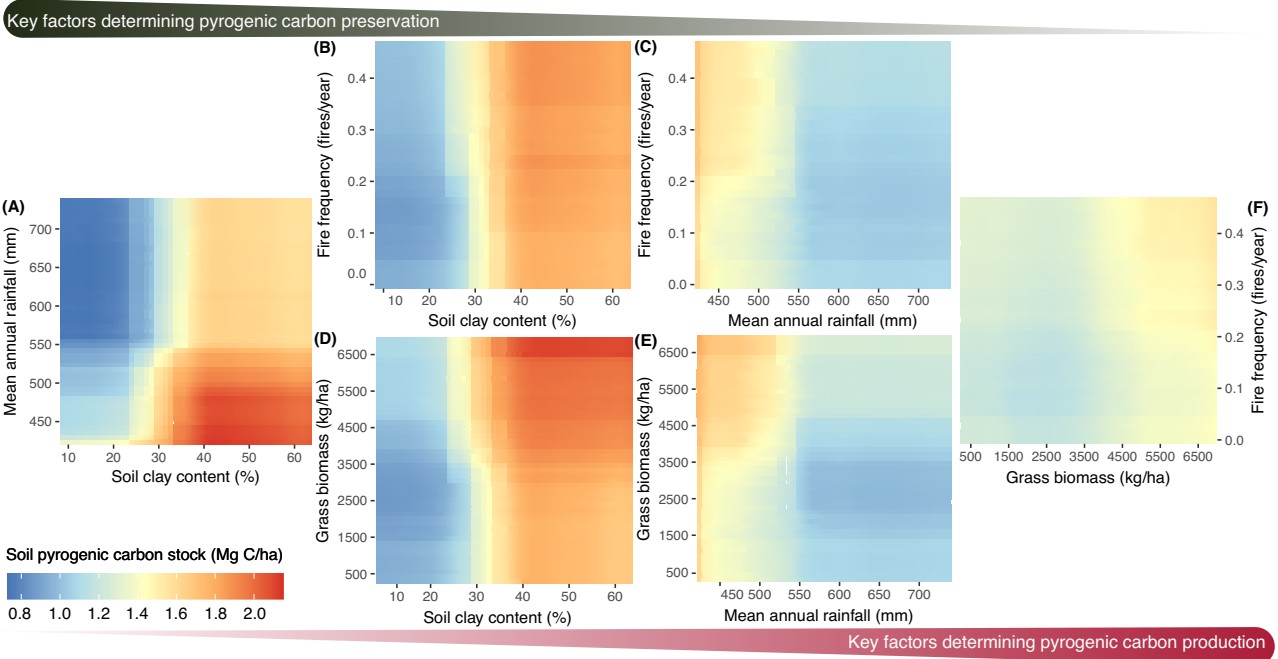

**Fig. 4 | Two-dimensional partial dependence plots illustrating the predicted soil pyrogenic carbon (PyC) stock (Mg C/ha) as a function of interactions among key predictor variables that influence PyC production and preservation.** The plot shows the effects of interactions between: (**A**) grass biomass and fire frequency, (**B**) soil clay and silt content and fire frequency, (**C**) mean annual rainfall and fire frequency, (**D**) soil clay and silt content and grass biomass, (**E**) mean annual rainfall and grass biomass, and (**F**) soil clay and silt content and mean annual rainfall on soil PyC stock across Kruger National Park, South Africa. Source data are provided as a Source Data file.

expected to persist longer in forest ecosystems[18]. These differences highlight that mechanisms of PyC preservation may matter consistently across ecosystems, but the extent that production matters may differ between forests and savannas, which have markedly different fire regimes and environmental conditions.

## Limitations and research priorities

While we find that soil texture and rainfall are key factors in the steady-state patterns of PyC accumulation, our study does not explicitly account for the fluxes of PyC into and out of soils, which are essential turnover processes and mechanisms linking production and preservation. It remains unknown, for example, the conversion rate of grass biomass to PyC and the factors influencing this conversion during fire events (but also see references[30,38,39]), the influence of fire severity on the quantity and quality of PyC in savannas (but see studies for forests[18,27]), the percentage of fire produced PyC integrated into soils over time, and the fate of the remaining PyC lost from ecosystems. Additionally, PyC consists of materials with varying stability and turnover rates[8,10,40], and labile PyC pools (e.g., pyro-sugars) are degradable by soil microbes over similar timescales as other carbon forms in SOC[41–44]. The proportion of degradable PyC and the role of biotic factors in PyC degradation and preservation in savanna soils remain unknown. We do note that while soil PyC stock is correlated with SOC stock, the percentage of PyC in SOC is unrelated to SOC itself (Supplementary Fig. 7), potentially suggesting that processes controlling PyC accumulation may, to some extent, differ from those governing other carbon forms in soils. Nevertheless, quantitatively assessing PyC fluxes into and out of savanna soils, including the distribution of PyC across stability pools, and how their turnover rates are influenced by abiotic and biotic factors, remains a critical research frontier.

Our conclusions are also limited to the range of environmental conditions found in Kruger. Our mean annual rainfall spanned ~420 to

740 mm, which may provide analogs for many savannas across different continents, but our results may not apply to humid savannas. For example, the Cerrado in Brazil receives 800 to 1600 mm of rainfall, supporting different herbaceous species and productivity, higher woody cover, and distinct fire regimes compared to the mesic savannas examined in this study[45]. While our results showed minimal impact of rainfall on soil PyC stock once it exceeds 560 mm in Kruger savannas (Fig. 3C), it is unclear whether this relationship can be extrapolated to the Cerrado. To establish a quantitative and mechanistic understanding of PyC accumulation and turnover in savannas—while accounting for the nonlinear response of soil PyC stocks to explanatory variables (Fig. 3)—additional cross-continental sampling of diverse ecosystems and environmental gradients is needed. We also note that our study focused on the 0–5 cm soil depth, relevant to the timescale of factors examined here, but future studies should explore PyC accumulation throughout the entire soil profile. This includes examining both vertical and lateral movement of PyC within the soil, processes that are well recognized as key factors influencing the spatial distribution and persistence of PyC in soils[17,46–48].

Lastly, accurately quantifying PyC content in soils remains a long-standing and actively debated challenge in the research community, largely due to the heterogenous nature of PyC, which consists of a wide range of materials with varying chemical structure and stability[10]. Each established method has its own strengths and limitations, and generally targets different fractions of the PyC continuum[25,26,49,50], making measurements difficult to compare across studies, as highlighted in our synthesis (Fig. 1). In this study, we used the peroxide/weak nitric acid digestion method, which is operationally practical for processing large sample sets with reasonable cost and resource requirements. However, this approach has limited specificity and may retain other oxidation-resistant organic materials in soils that can be misidentified as PyC[26]. Therefore, we suggest interpreting the absolute PyC values with caution and viewing our findings in a relative context, with an

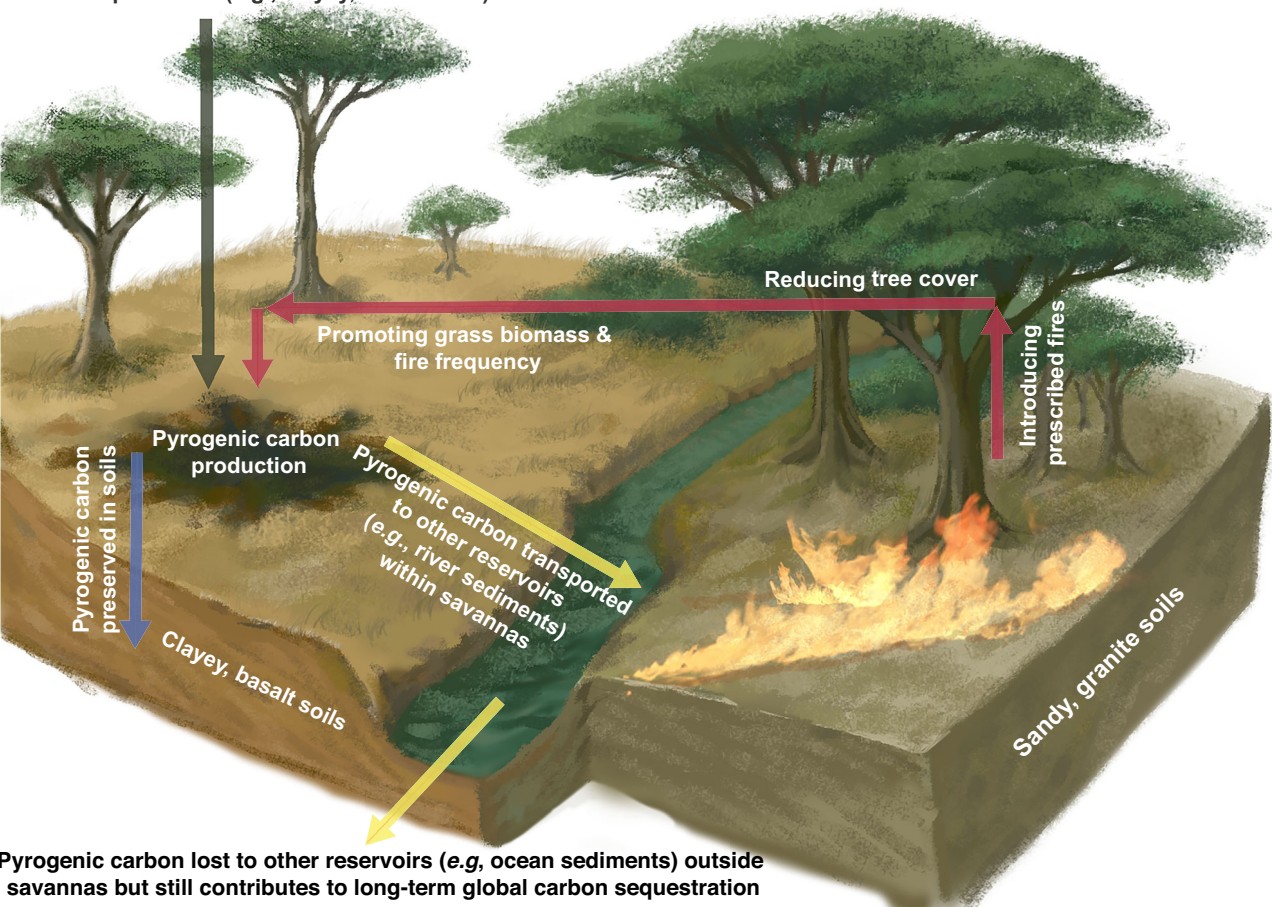

**Fig. 5 | Fire management to enhance the contribution of pyrogenic carbon (PyC) to savanna and global long-term carbon sequestration.** Fire management can prioritize savannas with higher potential for PyC preservation—such as those on clay-rich, basalt-derived soils—to enhance long-term carbon sequestration within these ecosystems. However, to maximize the contribution of savanna-derived PyC to global long-term carbon sequestration, fire management should also emphasize the use of prescribed burns to maintain savanna openness and restore natural fire regimes in savannas that have experienced historical woody encroachment.

emphasis on patterns and potential drivers of PyC variation rather than precise quantities.

### Implications for carbon dynamics across tropical savannas

The net effect of fire on ecosystem carbon balance largely depends on the timescale of the disturbance-recovery cycle. Unlike forest ecosystems, where post-fire vegetation recovery can take years to decades, depending on fire severity[51,52], savanna fires primarily burn the herbaceous layer but not the trees. The herbaceous layer can fully recover within the following growing season[53] and some species (e.g., forbs) even allocate more carbon and resources to belowground structure in response to fires[54]. Consequently, the carbon released through plant biomass combustion during the fire event can be recaptured by the new-growth biomass in savannas in a short timescale. As a result, the production and deposition of PyC through recurring fires represent a crucial net sink for carbon sequestration in savannas. Our results highlight that factors influencing PyC preservation outweigh those influencing PyC production in determining soil PyC accumulation, and most of these preservation factors, such as soil clay content and elevation, are relatively static. Focusing protection and management efforts on savanna regions with these characteristics may help enhance

savanna carbon sequestration (Fig. 5). Although fine-scale soil texture data may be less applicable for such efforts, landscape-scale parent material characteristics can effectively guide management. For example, Kruger is dominated by two major soil parent materials, granite and basalt. Basalt soils are clayey and nutrient-rich, supporting higher grass biomass and fire frequency, which in turn promote greater PyC production and preservation, leading to higher soil PyC stocks (Supplementary Fig. 8). Indeed, basalt covers only 34% of Kruger (1.90 million ha), yet it contributes approximately 51% of the total PyC stored in Kruger's surface soils (2.14 million Mg C), as estimated by our model. Therefore, fire management strategies can prioritize these basalt-dominated regions to enhance long-term carbon sequestration in savannas.

More importantly, the outsized role of savanna PyC in the global carbon cycle underscores the importance of conserving savanna ecosystems and maintaining their fire regimes (Fig. 5). In the past few decades, however, savanna burned areas, number of fires, and fire sizes have all declined[55], which has been attributed to agricultural expansion and intensification[55,56], as well as extensive woody encroachment[57,58]. Increasing tree cover reduces grass biomass and fire frequency (Supplementary Fig. 6), potentially weakening the pathway

of savanna carbon sequestration through PyC deposition from recurring fires. Active fire management, such as the application of prescribed burns in savannas prone to woody encroachment—like those on sandy granite soils with higher rainfall in Kruger—can be an effective tool for controlling tree cover and restoring natural fire return intervals[59–61]. Therefore, maintaining the open structure of savannas and their characteristic fire regimes is crucial not only for conserving savanna ecosystems but also for sustaining the production of PyC, which is central to the savanna carbon cycle and fire's role in the Earth system.

## Methods

### Global distribution of soil pyrogenic carbon measurements

In this study, we first reviewed the literature to compile a global dataset on soil PyC measurements. We searched the Web of Science (last accessed December 22, 2024) using keywords: "black carbon" OR "charcoal" OR "pyrogenic carbon" OR "fire-derived carbon" OR "pyrogenic organic matter" AND "Soil". Focusing on soil PyC measurements in natural ecosystems, we excluded agricultural and urban lands under extensive management and disturbance, and cases where PyC was intentionally added as a soil amendment (e.g., biochar). We also excluded PyC measurements if the analytical methods were not clear or if the measurements were not quantified in relation to total carbon content—for example, studies that reported only the amount of charcoal produced by fires. We also excluded studies that measured PyC only in the forest floor or organic layer and not in mineral soils. Since various analytical methods have been widely used to estimate PyC, making direct comparisons of absolute soil PyC concentrations or stocks challenging, we included only studies that reported the percentage of PyC in SOC or provided enough data to calculate this value based on reported PyC content/stock and total SOC content/stock. Overall, using these criteria, we collected 3456 PyC measurements from varying soil depths at 1139 unique locations across global natural ecosystems from 94 peer-review articles.

Additionally, we extracted geographic coordinates, mean annual precipitation and temperature, ecosystem type, fire regimes, sampling depth, soil texture, soil bulk density, and soil pH when available. If mean annual precipitation and temperature were not reported, we obtained them from WorldClim2.0 using site coordinates[62]. We also recorded the analytical methods used to measure PyC in soils, which include: benzenepolycarboxylic acid (BPCA), chemothermal oxidation, dichromate oxidation, nuclear magnetic resonance (NMR) spectroscopy, peroxide/weak nitric acid, and others (e.g., hydrogen pyrolysis, density fractionation, char separation, etc.).

### Soil pyrogenic carbon measurements across Kruger National Park, South Africa

Given the limited measurements of soil PyC in tropical and subtropical savannas, the overarching goal of this study is to evaluate the variation and drivers of soil PyC accumulation across these ecosystems. To achieve this, we conducted extensive spatial sampling across Kruger National Park (22°20′–25°30′S, 31°10′–32°00′E, hereafter Kruger), South Africa (Fig. 2A). Kruger is one of the largest protected areas across Africa, covering nearly 20,000 km² of subtropical and tropical savannas. A major advantage of this regional-scale study is the extensive research conducted in Kruger savannas, including long-term monitoring of fire regimes, vegetation dynamics, and environmental conditions. These comprehensive datasets provide a strong foundation for evaluating the drivers of soil PyC accumulation across tropical and subtropical savannas.

We used 253 Vegetation Condition Assessment (VCA) sites across Kruger for this study (Fig. 2B). Starting from 1989, Kruger management established 533 VCA sites to monitor grass biomass to inform fire management. Grass biomass was measured annually between March and April in 50 m × 60 m plots using a calibrated disc pasture meter[63].

Measurements were taken every 2 m along four 50-m transects positioned at 0 m, 20 m, 40 m, and 60 m along the plot's length, resulting in a total of 204-disc pasture meter readings per plot to calculate average plot-level grass biomass. Biomass measurements were conducted yearly from 1989 to 2012 and again in 2016 and 2021. However, the number and identity of plots sampled each year varied substantially (ranging from 111 to 532 sites, with mean = 382 sites, 25th percentile = 205 sites, and 75th percentile = 507 sites) (Supplementary Fig. 9). The average grass biomass over 30 years (from 1989 to 2012 and 2016 to 2021) was calculated to represent the long-term grass biomass across 253 sampled VCA sites.

At each of these 253 VCA sites, we identified the center of the 50 m × 60 m plot. Six sampling points were selected, radiating 10 meters away from the center at approximately 60-degree intervals. At each sampling point, surface litter (if present) was removed, and soil samples from 0–5 cm depth were collected using a soil core (5 cm in diameter and 5 cm in length) during August to October 2023. These six samples were then combined to form a composite sample for the site. Soil samples were air-dried and passed through a 2-mm sieve. Soil bulk density for each site was estimated by oven-drying a known volume of soil. Pyrogenic carbon inside each soil sample was determined using a peroxide/weak nitric acid digestion method[26]. Briefly, a 1.000 g sample of oven-dry soil was weighed into a glass tube and treated with 20 mL of 30% $H_2O_2$ and 10 mL of 1 M $HNO_3$. The tube was gently swirled at room temperature for 30 min, then heated to 100 °C on a heating plate for 16 h. During this period, the sample was swirled occasionally, and effervescence was monitored. If effervescence persisted after 16 h, heating continued until no further effervescence occurred, indicating that digestion was complete. The soil sample was then filtered through Whatman #2 filter paper, oven-dried at 65 °C, and the soil recovery weight was determined. Oven-dried soils were homogenized using a mortar and pestle, and total carbon was determined using a Costech ECS 4010 elemental analyzer (Costech Analytical Tech Inc., Valencia, CA). The total carbon measured after digestion, accounting for weight loss, is reported as total PyC, assuming all non-PyC was consumed during the digestion process. Total carbon in undigested soil samples was also analyzed with the elemental analyzer and reported as total soil organic carbon (SOC). The total soil PyC stock (Mg C/ha) within the 0–5 cm soil depth was calculated based on soil bulk density and depth. Additionally, the percentage of PyC in SOC was calculated by dividing the amount of PyC by the total SOC.

Kruger maintains continuous records of the spatial distribution of fires dating back to 1941. Using these records, we extracted fire frequency (fires/year) and years since the last fire for each VCA site. Fire frequency for 253 VCA sites ranged from 0 to 0.46 fires/year between 1941 and 2023, with an average of 0.20 fires/year (n = 253) (Supplementary Fig. 4). Years since the last fire for these sites ranged from 0 to 83 years, with an average of 8.9 years (n = 253) (Supplementary Fig. 4). We also considered woody cover, as evidence suggests that increasing woody cover reduces both grass biomass and fire frequency[64,65]. Woody cover for each VCA site was extracted from a remote sensing product (30 m resolution) for Kruger based on the European Space Agency's Copernicus Sentinel-1 radar satellite and calibrated and validated using high-resolution light detection and ranging data[66]. Woody cover for the 253 VCA sites ranged from 1.0 to 65.1%, with an average of 18.9% (n = 253) (Supplementary Fig. 4).

Kruger has continuously collected rainfall data through a network of up to 22 stations across the entire park since the 1960s. Long-term data from these stations were used to interpolate a mean annual rainfall map for Kruger (Supplementary Fig. 4). Mean annual rainfall values for the 253 VCA sites were then extracted from this map, ranging from 424 to 737 mm, with an average of 537 mm (n = 253). A survey of geomorphic and soil characteristics in Kruger was conducted between 1986 and 1989. Rapid estimates of soil clay content (%) for the A horizon (primarily 0–20 cm depth) were obtained at 1794 sites using

the feel method. These estimates were then correlated and adjusted based on soil clay content determined through particle-size analysis of representative soil samples[67]. The measurements were used to interpolate a soil clay content map for Kruger (Supplementary Fig. 4C). Soil clay content values for the VCA sites were extracted from this map, ranging from 8.6% to 63.3%, with an average of 26.8% ($n = 253$) (Supplementary Fig. 4). Elevations for the 253 VCA sites were obtained from the NASA Digital Elevation Model (NASADEM) at 30 m resolution, generated using Shuttle Radar Topography Mission data. Site elevations ranged from 179 to 638 m above sea level, with an average of 339 m ($n = 253$). A slope map for Kruger was also derived from NASADEM, and slope values for the VCA sites were extracted, ranging from 0.23° to 8.97°, with an average of 2.15° ($n = 253$) (Supplementary Fig. 4).

### Data analysis

We used a random forest (RF) machine learning algorithm to assess the relative importance of eight explanatory factors in predicting soil PyC stocks (Mg C/ha) across savannas in Kruger. Our eight explanatory factors can be broadly categorized into those influencing PyC production –grass biomass, fire frequency, years since the last fire, and woody cover– and those affecting PyC preservation in soils, including soil clay content, mean annual rainfall, elevation, and slope. The primary reason for choosing an RF model is its ability to capture nonlinear relationships between predictors and response variables[68], which is essential for representing the complex interactions in this regional-scale analysis. Prior to building the RF model, we checked the correlation coefficient matrix among the predictors to ensure that multicollinearity was not present in the data (Supplementary Fig. 10). In this study, we used an iterative approach to optimize the model for PyC stock by adjusting the total number of trees ($n_{tree}$) and the number of predictor variables randomly selected at each split ($m_{try}$). Model performance was assessed using 10-fold cross-validation, with root mean squared error (RMSE) and the percentage of variance explained ($R^2$) as evaluation metrics. Based on these metrics, the final model parameters were set to 700 for $n_{tree}$ and 3 for $m_{try}$. The optimization process was conducted using the "caret" package in R, followed by RF analysis using the "randomForest" package. Additionally, Spatial bias in the model predictions was assessed by mapping the residuals from the RF and visually inspecting their spatial patterning (Supplementary Fig. 5), as well as calculating Moran's I of the residuals. We also assessed the performance of a generalized linear model (GLM) for predicting soil PyC stocks and selected the preferred model based on Akaike's Information Criterion (AIC) (Supplementary Table 1). Compared to the preferred GLM model ($R^2 = 0.60$, RMSE), the RF model ($R^2 = 0.67$, RMSE = 0.44 Mg C/ha) demonstrated improved performance by incorporating nonlinear relationships between the response and explanatory variables.

After selecting the optimized RF model, we evaluated the relative importance of each predictor variable on soil PyC stock by measuring the increase in the validation dataset's mean square error (MSE) when each variable was sequentially removed from the model. Partial dependence plots (PDPs) were then used to examine the relationship between soil PyC stock and each predictor variable. These plots illustrate the modeled soil PyC stock over the range of each predictor variable while holding the remaining variables constant across the dataset. To assess the effects of two-way interactions between predictor variables on soil PyC stock, we extended this approach by systematically modifying two predictor variables simultaneously. We focused on interactions between grass biomass and fire frequency, which are key drivers of PyC production, as well as between mean annual rainfall and soil clay content, which are critical factors for PyC preservation in soils.

## Data availability

The data generated in this study have been deposited in the Zenodo repository and are freely available for download at: https://doi.org/10.5281/zenodo.15936377; Source data are provided with this paper.

## Code availability

The code used in this study has been deposited in the Zenodo repository and is available for download at: https://doi.org/10.5281/zenodo.15936377.

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

## Acknowledgments

We thank Noel Nzima for assisting with soil sample collection, Ayla Meek for lab analysis of soil pyrogenic carbon, SANParks for granting permission to work in Kruger National Park (project number: SS1218), Ying Huang for creating the artwork in Fig. 5, and Dr. Tongxin Hu (Northeast Forestry University, China) for contributing pyrogenic carbon measurements to the global synthesis. This work was supported by the Research Catalyst Program at Utah State University and by startup funds from the University of California, Santa Barbara, to Y.Z.

## Author contributions

Y.Z. and C.C. designed the study; C.C. coordinated the field sampling; Y.Z. collected data from the literature and conducted the laboratory analysis; Y.Z. analyzed the data and wrote the first draft of the manuscript; Y.Z., A.T.K., A.S. and C.C. reviewed and edited the manuscript. All authors approved the final submission.

## Competing interests

The authors declare no competing interests.
