## [Peer Review File · Nature Communications]

Pyrogenic carbon contribution to tropical savanna soil carbon storage

Corresponding Author: Dr Yong Zhou

Version 0:

Reviewer comments:

Reviewer #1

(Remarks to the Author)

Key Results:

This manuscript describes and quantifies the pyrogenic carbon concentrations across savannas in the Kruger National Park, South Africa. Soil PyC stocks were positively correlated with clay content and negatively correlated with rainfall. Higher PyC stocks were also associated with increased grass density and more frequent fire. This work also includes a meta-analysis of global PyC stocks on natural systems, which identified the African continent as underrepresented in soil PyC studies.

Validity and Significance:

This work measures soil PyC in an understudied region, which is an important area to improve our understanding of the global soil PyC cycle. However, there are some significant weaknesses to this work that the authors have not addressed.

Data and methodology/Analytical approach

In the meta-analysis, the results are nicely laid out in Figure 1, but I feel that there are more interesting questions that could be answered – were there higher soil PyC concentrations across different biomes? How do global temperatures and precipitation dictate PyC concentrations? This would help directly link the meta-analysis to the results from Kruger National Park. This could perhaps be achieved by subsetting the data by analysis method to compare across sites.

In the field portion of this study, the KMD method is used to quantify soil PyC. This method is not the best accepted method for quantifying soil PyC and ideally should be paired with a more directly quantitative method (e.g., BPCA, NMR) that is more specific for PyC. Since this method is a digestion, it can leave behind materials that are not PyC, but simply resistant to oxidation, and can vary across soil textures and parent materials. These results need to be carefully interpreted in order to ensure that they are not overly interpreted as very quantitative. This method is more appropriate for a study wherein PyC is measured on more similar soil types to indicate increase/decreasing concentrations of PyC, instead of as a direct quantitative measure.

Minor comments for suggested improvements:

Abstract and elsewhere

- Folks in the soil science community have widely abandoned the term “recalcitrant” as it is not specific and used to mean two different things – either persistent/stable carbon or chemically/kinetically difficult to break down. It is not clear here what the authors mean by this term, as they describe later in the manuscript that rapid breakdown of PyC is observed. See discussions in the following references:

Kleber 2010 Environ. Chem. What is recalcitrant soil organic matter?

Lehmann and Kleber. 2015 Nature. The contentious nature of soil organic matter.

Schmidt et al., 2011 Nature. Persistence of soil organic matter as an ecosystem property.

Final sentence of the abstract – The results of this work do not directly address the stability of PyC or its role of PyC in carbon sequestration (but they have included proportion of stocks). I suggest revising this to reflect the scope of this work.

Clarity/context/references:

The work is clearly laid out and easy to follow. However, there could be more citations of previous work, for example, in the loss processes of PyC, there is relatively little mention of decomposition or leaching of PyC, each of which have relatively broad bodies of work investigating these processes.

Reviewer #2

(Remarks to the Author)

The statement made in the abstract that the magnitude of the PyC contribution to soil organic carbon (SOC) storage in savannas remain unknown does not seem well justified. Since the reported estimate of the PyC contribution is close to the estimate reported by Saiz et al. (2015), this statement should be revised. The PyC contribution is not unknown, there are some estimates in literature, and therefore this study checks the validity of existing estimates and reports an empirical model attributing the PyC contribution to SOC.

The derived model cannot be used outside the range of the observed values of the drivers that were used to develop this model (a discussion of the use of empirical models can be found, for example, in the paper, When does artificial intelligence replace process-based models in ecological modelling? <https://doi.org/10.1016/j.ecolmodel.2024.110923>). Nevertheless, it is important to make the derived model available for those who might be interested in predicting PyC in African savannas based on the study in Kruger National Park.

I did not find any obvious flaws that may prohibit publication of the reported results, which are sufficiently significant for publication in NCOMMS.

Version 1:

Reviewer comments:

Reviewer #1

(Remarks to the Author)

The authors have submitted a revised version of this manuscript, which describes PyC stocks in a meta-analysis and across Kruger National Park. This revised manuscript has responded to reviewer comments thoroughly. I have added a few minor comments below that should be addressed.

Abstract – It would be good to mention the KMD method as how PyC was determined in the abstract.

Line 115-6: What is the total % of the world that is covered by tropical savanna? This would be helpful to illustrate how underrepresented this ecosystem is for soil PyC measurements

Line 171-3: Pyrogenic carbon should be abbreviated here to be consistent.

Line 188: It is not clear what is mean by “labile to stable PyC” as a proportion. This could be edited for clarity – I would assume that the authors are indicating the amount of PyC that could be quickly decomposed, but they also could be referring to pyro-sugars.

Line 224-5: The net effect also depends on burn severity and fire regime.

Line 236-8: This sentence needs editing for clarity.

Reviewer #2

(Remarks to the Author)

I think the revised version of the manuscript is acceptable in its present form.

Response Letter

Title: Pyrogenic carbon contribution to tropical savanna soil carbon storage

Manuscript ID: NCOMMS-25-34357-T

Reviewer #1 (Remarks to the Author):

Key Results:

This manuscript describes and quantifies the pyrogenic carbon concentrations across savannas in the Kruger National Park, South Africa. Soil PyC stocks were positively correlated with clay content and negatively correlated with rainfall. Higher PyC stocks were also associated with increased grass density and more frequent fire. This work also includes a meta-analysis of global PyC stocks on natural systems, which identified the African continent as underrepresented in soil PyC studies.

Validity and Significance:

This work measures soil PyC in an understudied region, which is an important area to improve our understanding of the global soil PyC cycle. However, there are some significant weaknesses to this work that the authors have not addressed.

Response: Thank you for your positive feedback on our manuscript and for recognizing the importance of studying soil PyC in this understudied savanna region. We appreciate your constructive comments and have addressed each of the identified weaknesses in detail in our revised manuscript. Please see our point-by-point responses below.

Data and methodology/Analytical approach

In the meta-analysis, the results are nicely laid out in Figure 1, but I feel that that more interesting questions could be answered – were there higher soil PyC concentrations across different biomes? How do global temperatures and precipitation dictate PyC concentrations? This would help directly link the meta-analysis to the results from Kruger National Park. This could perhaps be achieved by subsetting the data by analysis method to compare across sites.

Response: Thank you for this thoughtful comment. In our revised manuscript, we expanded our analysis to examine the percentage of PyC in SOC in relation to mean annual temperature (MAT), mean annual precipitation (MAP), ecosystem type by different analytical method used to quantify PyC, and soil depth. These descriptive comparisons offer insight into potential patterns across global natural ecosystems and soil profiles. However, we did not conduct formal statistical analyses on these patterns due to the substantial uncertainty introduced by methodological inconsistencies across the studies. Please refer to Lines 94-110 and supplementary figs. 1-3.

To promote further use of this global synthesized data, we have uploaded the full dataset to an open-access repository. This enables readers to explore the data in more detail, subset it based on their research questions, and apply analytical approaches most appropriate to their objectives.

In the field portion of this study, the KMD method is used to quantify soil PyC. This method is not the best accepted method for quantifying soil PyC and ideally should be paired with a more directly quantitative method (e.g., BPCA, NMR) that is more specific for PyC. Since this method is a digestion, it can leave behind materials that are not PyC, but simply resistant to oxidation, and can vary across soil textures and parent materials. These results need to be carefully interpreted in order to ensure that they are not overly interpreted as very quantitative. This method is more appropriate for a study wherein PyC is measured on more similar soil types to indicate increase/decreasing concentrations of PyC, instead of as a direct quantitative measure.

*Response: We agree that accurately quantifying soil PyC is a long-standing and actively debated challenge in the PyC research community. As the reviewer rightly points out, each available method for PyC quantification has its own strengths and limitations, and no single approach is universally accepted as definitive at present. This is also evident in our synthesis: among the 3,456 measurements we compiled from the literature, a wide range of methods were used, including benzenepolycarboxylic acid (BPCA, n = 815), chemothermal oxidation (n = 637), dichromate oxidation (n = 593), nuclear magnetic resonance (NMR) spectroscopy (n = 585), peroxide/weak nitric acid digestion (n = 414), and various other approaches (n = 502). Even the two methods suggested by the reviewer—BPCA and NMR—have notable limitations. The BPCA method is widely used for its specificity in detecting condensed polycyclic structures typical of PyC, but it requires harsh acid digestion that destroys much of the original structure, preventing full recovery of PyC as BPCAs. Although conversion factors have been proposed to estimate total PyC from BPCA yields, their accuracy is limited by the heterogeneous aromatic condensation of most chars (Cotrufo et al. 2016; Wiedemeier et al. 2016). NMR spectroscopy offers detailed molecular-level information and can distinguish PyC from other forms of soil organic matter without chemical alteration. However, it is expensive, instrument-intensive, and requires relatively high PyC concentrations for reliable detection, which can be problematic in low-carbon soils. In addition, this method often yields higher PyC estimates due to its broader detection of aromatic materials, which may not always align with more conservative or operational definitions of PyC (Bornemann et al. 2008). This study is not intended to resolve the methodological challenges surrounding PyC quantification. Rather, our goal was to synthesize existing data to highlight the inconsistencies in PyC measurements arising from different approaches, and then apply a single method to examine factors driving PyC accumulation in savanna soils. We chose the peroxide/weak nitric acid digestion method due to its practicality for processing a large number of samples with reasonable costs and resources. We acknowledge the limitations of this method and have clearly highlighted these caveats in the revised manuscript. We caution readers against overinterpreting the absolute values of PyC and emphasize that our results should be interpreted in a relative, rather than strictly quantitative, context. Please refer to **Lines 211-222** in the revised manuscript.*

*Wiedemeier, D. B., Lang, S. Q., Gierga, M., Abiven, S., Bernasconi, S. M., Früh-Green, G. L., ... & Schmidt, M. W. (2016). Characterization, quantification and compound-specific isotopic analysis of pyrogenic carbon using benzene polycarboxylic acids (BPCA). *Journal of Visualized Experiments: JoVE*, (111), 53922.*

*Cotrufo, M. F., Boot, C., Abiven, S., Foster, E. J., Haddix, M., Reisser, M., ... & Schmidt, M. W. (2016). Quantification of pyrogenic carbon in the environment: An integration of analytical approaches. *Organic Geochemistry*, 100, 42-50.*

Bornemann, L., Welp, G., Brodowski, S., Rodionov, A., & Amelung, W. (2008). Rapid assessment of black carbon in soil organic matter using mid-infrared spectroscopy. Organic Geochemistry, 39(11), 1537-1544.

Minor comments for suggested improvements:

Abstract and elsewhere

Folks in the soil science community have widely abandoned the term “recalcitrant” as it is not specific and used to mean two different things – either persistent/stable carbon or chemically/kinetically difficult to break down. It is not clear here what the authors mean by this term, as they describe later in the manuscript that rapid breakdown of PyC is observed. See discussions in the following references:

Kleber 2010 Environ. Chem. What is recalcitrant soil organic matter?

Lehmann and Kleber. 2015 Nature. The contentious nature of soil organic matter.

Schmidt et al., 2011 Nature. Persistence of soil organic matter as an ecosystem property.

Response Thank you for pointing this out. We agree that the term “recalcitrant” is problematic due to its ambiguity, as it has been used to describe both chemically resistant compounds and carbon with long residence times in soils. In our manuscript, we intended to refer to the potential for PyC to persist in soils over long timescales. To improve clarity and align with current terminology in the soil science literature, we have replaced “recalcitrant” with “persistent”. We have also revised relevant sections to distinguish between persistence and degradation dynamics more clearly.

Final sentence of the abstract – The results of this work do not directly address the stability of PyC or its role of PyC in carbon sequestration (but they have included proportion of stocks). I suggest revising this to reflect the scope of this work.

Response: We agree that our study does not directly assess the long-term stability of PyC or quantify its role in carbon sequestration. To better reflect the scope of our work, we have revised the final sentence of the abstract to focus on the empirical findings regarding PyC distribution and its environmental drivers, and to clarify that our results improve our understanding of fires’ role in the tropical savanna carbon cycle. Please refer to lines 36-38.

Clarity/context/references:

The work is clearly laid out and easy to follow. However, there could be more citations of previous work, for example, in the loss processes of PyC, there is relatively little mention of decomposition or leaching of PyC, each of which have relatively broad bodies of work investigating these processes.

Response: Thank you for your thoughtful feedback and positive remarks about the clarity of our work. We agree that decomposition and transportation processes are key to the accumulation of PyC in soils. The decomposition processes are briefly addressed in our manuscript (please refer to lines 186-188), and in the revised version, we emphasized transportation processes and included additional relevant literature (please refer to lines 208-210). However, since our study did not directly investigate these processes, we have limited our interpretations to what can be supported by our results, without extensively extending the discussion to decomposition and

transportation mechanisms. We welcome any more specific suggestions from the reviewer and will incorporate additional detail as appropriate.

Reviewer #2 (Remarks to the Author):

The statement made in the abstract that the magnitude of the PyC contribution to soil organic carbon (SOC) storage in savannas remain unknown does not seem well justified. Since the reported estimate of the PyC contribution is close to the estimate reported by Saiz et al. (2015), this statement should be revised. The PyC contribution is not unknown, there are some estimates in literature, and therefore this study checks the validity of existing estimates and reports an empirical model attributing the PyC contribution to SOC.

Response: Our intention in the abstract was to highlight that while some estimates exist, most measurements to date have been conducted in forest ecosystems in the Northern Hemisphere. We agree that previous studies, such as Saiz et al. (2015), have provided valuable estimates of PyC contributions to SOC, particularly in Australian savannas. However, there remains limited data from savannas, especially regarding the factors that drive variability in PyC contributions to SOC in these systems. To address your concern, we have revised the abstract (Lines 26-27 in the revised manuscript) to reflect the current state of knowledge more accurately.

The derived model cannot be used outside the range of the observed values of the drivers that were used to develop this model (a discussion of the use of empirical models can be found, for example, in the paper, When does artificial intelligence replace process-based models in ecological modelling? <https://doi.org/10.1016/j.ecolmodel.2024.110923>). Nevertheless, it is important to make the derived model available for those who might be interested in predicting PyC in African savannas based on the study in Kruger National Park.

Response: Thank you for this helpful comment. We agree that empirical models, such as ours, must be interpreted and applied with caution. As noted in the “Limitations and Research Priorities” section of the manuscript, we emphasize that the derived model, based on field data from Kruger National Park, is only valid within the range of predictor variables used to develop it. To address this limitation transparently, we have clearly stated that the model should not be extrapolated beyond the conditions observed in our study (please refer to Lines 196-206 in the revised manuscript).

In the main text, we present results based on a random forest model, which captures potential non-linear relationships between PyC and environmental drivers. Additionally, to enhance reproducibility and accessibility, we also included a generalized linear regression model for comparison (Lines 136-139). The full model specifications, including parameter estimates and predictor ranges, are provided in the supplementary materials (please refer to Supplementary Table 1). These details allow readers to explore the model further and, if appropriate, apply it to other African savanna sites with similar environmental conditions. Additionally, we have uploaded our full dataset and code to an open-source repository. Readers can download the data and model specifications to reproduce our analyses or apply the models to their own study systems, provided the environmental conditions fall within the range of our observed data.

While we acknowledge the limitations of empirical modeling, we believe our work provides a valuable step toward understanding and predicting PyC contributions to SOC in fire-prone

savanna ecosystems. The derived models offer both conceptual and practical tools for researchers and land managers working in similar contexts.

I did not find any obvious flaws that may prohibit publication of the reported results, which are sufficiently significant for publication in NCOMMS.

Response: Thank you for your positive assessment of our work and for recognizing the significance of our findings. We have made minor revisions to further clarify our key points and strengthen the manuscript.